# DeepPINK: reproducible feature selection in deep neural networks

**Yang Young Lu** [*]
Department of Genome Sciences
University of Washington
Seattle, WA 98195
ylu465@uw.edu

**Yingying Fan** [*]
Data Sciences and Operations Department
Marshall School of Business
University of Southern California
Los Angeles, CA 90089
fanyingy@marshall.usc.edu

**Jinchi Lv**
Data Sciences and Operations Department
Marshall School of Business
University of Southern California
Los Angeles, CA 90089
jinchilv@marshall.usc.edu

**William Stafford Noble**
Department of Genome Sciences and Department of Computer Science and Engineering
University of Washington
Seattle, WA 98195
william-noble@uw.edu

## Abstract

Deep learning has become increasingly popular in both supervised and unsupervised machine learning thanks to its outstanding empirical performance. However, because of their intrinsic complexity, most deep learning methods are largely treated as black box tools with little interpretability. Even though recent attempts have been made to facilitate the interpretability of deep neural networks (DNNs), existing methods are susceptible to noise and lack of robustness. Therefore, scientists are justifiably cautious about the reproducibility of the discoveries, which is often related to the interpretability of the underlying statistical models. In this paper, we describe a method to increase the interpretability and reproducibility of DNNs by incorporating the idea of feature selection with controlled error rate. By designing a new DNN architecture and integrating it with the recently proposed knockoffs framework, we perform feature selection with a controlled error rate, while maintaining high power. This new method, DeepPINK (Deep feature selection using Paired-Input Nonlinear Knockoffs), is applied to both simulated and real data sets to demonstrate its empirical utility. [2]

## 1   Introduction

Rapid advances in machine learning techniques have revolutionized our everyday lives and had profound impacts on many contemporary domains such as decision making, healthcare, and finance

---

[*]These authors contributed equally to this work.

[2] All code and data will be available here: github.com/younglululu/DeepPINK.

[28]. In particular, deep learning has received much attention in recent years thanks to its outstanding empirical performance in both supervised and unsupervised machine learning tasks. However, due to the complicated nature of deep neural networks (DNNs) and other deep learning methods, they have been mostly treated as black box tools. In many scientific areas, interpretability of scientific results is becoming increasingly important, as researchers aim to understand why and how a machine learning system makes a certain decision. For example, if a clinical image is predicted to be either benign or malignant, then the doctors are eager to know which parts of the image drive such a prediction. Analogously, if a financial transaction is flagged to be fraudulent, then the security teams want to know which activities or behaviors led to the flagging. Therefore, an explainable and interpretable system to reason about why certain decisions are made is critical to convince domain experts [27].

The interpretability of conventional machine learning models, such as linear regression, random forests, and support vector machines, has been studied for decades. Recently, identifying explainable features that contribute the most to DNN predictions has received much attention. Existing methods either fit a simple model in the local region around the input [33, 38] or locally perturb the input to see how the prediction changes [35, 6, 34, 37]. Though these methods can yield insightful interpretations, they focus on specific architectures of DNNs and can be difficult to generalize. Worse still, Ghorbani *et al.* [22] systematically revealed the fragility of these widely-used methods and demonstrated that even small random perturbation can dramatically change the feature importance. For example, if a particular region of a clinical image is highlighted to explain a malignant classification, the doctors would then focus on that region for further investigation. However, it would be highly problematic if the choice of highlighted region varied dramatically in the presence of very small amounts of noise.

In such a setting, it is desirable for practitioners to select explanatory features in a fashion that is robust and reproducible, even in the presence of noise. Though considerable work has been devoted to creating feature selection methods that select relevant features, it is less common to carry out feature selection while explicitly controlling the associated error rate. Among different feature selection performance measures, the false discovery rate (FDR) [4] can be exploited to measure the performance of feature selection algorithms. Informally, the FDR is the expected proportion of falsely selected features among all selected features, where a false discovery is a feature that is selected but is not truly relevant (For a formal definition of FDR, see section 2.1). Commonly used procedures, such as the Benjamini–Hochberg (BHq) procedure [4], achieve FDR control by working with p-values computed against some null hypothesis, indicating observations that are less likely to be null.

In the feature selection setting, existing methods for FDR control utilize the p-values produced by an algorithm for evaluating feature importance, under the null hypothesis that the feature is not relevant. Specifically, for each feature, one tests the significance of the statistical association between the specific feature and the response either jointly or marginally and obtains a p-value. These p-values are then used to rank the feature importance for FDR control. However, for DNNs, how to produce meaningful p-values that can reflect feature importance is still completely unknown. See, e.g., [19] for the nonuniformity of p-values for the specific case of diverging-dimensional generalized linear models. Without a way to produce appropriate p-values, performing feature selection with a controlled error rate in deep learning is highly challenging.

To bypass the use of p-values but still achieve FDR control, Candès *et al.* [10] proposed the model-X knockoffs framework for feature selection with controlled FDR. The salient idea is to generate knockoff features that perfectly mimic the arbitrary dependence structure among the original features but are conditionally independent of the response given the original features. Then these knockoff features can be used as control in feature selection by comparing the feature importance between original features and their knockoff counterpart. See more details in section 2.2 for a review of the model-X knockoffs framework and [2, 3, 10] for more details on different knockoff filters and their theoretical guarantees.

In this paper, we integrate the idea of a knockoff filter with DNNs to enhance the interpretability of the learned network model. Through simulation studies, we discover surprisingly that naively combining the knockoffs idea with a multilayer perceptron (MLP) yields extremely low power in most cases (though FDR is still controlled). To resolve this issue, we propose a new DNN architecture named DeepPINK (Deep feature selection using Paired-Input Nonlinear Knockoffs). DeepPINK is built upon an MLP with the major distinction that it has a plugin pairwise-coupling layer containing $p$ filters, one per each input feature, where each filter connects the original feature and its knockoff counterpart. We demonstrate empirically that DeepPINK achieves FDR control with much higher

power than many state-of-the-art methods in the literature. We also apply DeepPINK to two real data sets to demonstrate its empirical utility. It is also worth mentioning that the idea of DeepPINK may be generalized to other deep neural networks such as CNNs and RNNs, which is the subject of our ongoing research.

## 2   Background

### 2.1   Model setting

Consider a supervised learning task where we have $n$ independent and identically distributed (i.i.d.) observations $(\mathbf{x}_i, Y_i)$, $i = 1, \cdots, n$, with $\mathbf{x}_i \in \mathbb{R}^p$ the feature vector and $Y_i$ the scalar response. Here we consider the high-dimensional setting where the feature dimensionality $p$ can be much larger than the sample size $n$. Assume that there exists a subset $\mathcal{S}_0 \subset \{1, \cdots, p\}$ such that, conditional on features in $\mathcal{S}_0$, the response $Y_i$ is independent of features in the complement $\mathcal{S}_0^c$. Our goal is to learn the dependence structure of $Y_i$ on $\mathbf{x}_i$ so that effective prediction can be made with the fitted model and meanwhile achieve accurate feature selection in the sense of identifying features in $\mathcal{S}_0$ with a controlled error rate.

### 2.2   False discovery rate control and the knockoff filter

To measure the accuracy of feature selection, various performance measures have been proposed. The false discovery rate (FDR) is among the most popular ones. For a set of features $\widehat{S}$ selected by some feature selection procedure, the FDR is defined as

$$\text{FDR} = \mathbb{E}[\text{FDP}] \text{ with FDP} = \frac{|\widehat{S} \cap \mathcal{S}_0^c|}{|\widehat{S}|},$$

where $|\cdot|$ stands for the cardinality of a set. Many methods have been proposed to achieve FDR control [5, 36, 1, 15, 16, 40, 14, 23, 20, 44, 17]. However, as discussed in Section 1, most of these existing methods rely on p-values and cannot be adapted to the setting of DNNs.

In this paper, we focus on the recently introduced model-X knockoffs framework [10]. The model-X knockoffs framework provides an elegant way to achieve FDR control in a feature selection setting at some target level $q$ in finite sample and with arbitrary dependency structure between the response and features. The idea of knockoff filters was originally proposed in Gaussian linear models [2, 3]. The model-X knockoffs framework generalizes the original method to work in arbitrary, nonlinear models. In brief, knockoff filter achieves FDR control in two steps: 1) construction of knockoff features, and 2) filtering using knockoff statistics.

**Definition 1** ([10])**.** Model-X knockoff features for the family of random features $\mathbf{x} = (X_1, \cdots, X_p)^T$ are a new family of random features $\tilde{\mathbf{x}} = (\tilde{X}_1, \cdots, \tilde{X}_p)^T$ that satisfies two properties: (1) $(\mathbf{x}, \tilde{\mathbf{x}})_{\text{swap}(\mathcal{S})} \overset{d}{=} (\mathbf{x}, \tilde{\mathbf{x}})$ for any subset $\mathcal{S} \subset \{1, \cdots, p\}$, where $\text{swap}(\mathcal{S})$ means swapping $X_j$ and $\tilde{X}_j$ for each $j \in \mathcal{S}$ and $\overset{d}{=}$ denotes equal in distribution, and (2) $\tilde{\mathbf{x}} \perp\!\!\!\perp Y | \mathbf{x}$, i.e., $\tilde{\mathbf{x}}$ is independent of response $Y$ given feature $\mathbf{x}$.

According to Definition 1, the construction of knockoffs is totally independent of the response $Y$. If we can construct a set of model-X knockoff features, then by comparing the original features with these control features, FDR can be controlled at target level $q$. See [10] for theoretical guarantees of FDR control with knockoff filters.

Clearly, the construction of model-X knockoff features plays a key role in FDR control. In some special cases such as $\mathbf{x} \sim \mathcal{N}(0, \boldsymbol{\Sigma})$ with $\boldsymbol{\Sigma} \in \mathbb{R}^{p \times p}$ the covariance matrix, the model-X knockoff features can be constructed easily. More specifically, if $\mathbf{x} \sim \mathcal{N}(0, \boldsymbol{\Sigma})$, then a valid construction of knockoff features is

$$\tilde{\mathbf{x}} | \mathbf{x} \sim N\left(\mathbf{x} - \text{diag}\{\mathbf{s}\}\boldsymbol{\Sigma}^{-1}\mathbf{x}, 2\text{diag}\{\mathbf{s}\} - \text{diag}\{\mathbf{s}\}\boldsymbol{\Sigma}^{-1}\text{diag}\{\mathbf{s}\}\right). \tag{1}$$

Here diag $\{\mathbf{s}\}$ with all components of $\mathbf{s}$ being positive is a diagonal matrix the requirement that the conditional covariance matrix in Equation 1 is positive definite. Following the above knockoffs

construction, the original features and the model-X knockoff features have the following joint distribution

$$(\mathbf{x}, \tilde{\mathbf{x}}) \sim \mathcal{N}\left(\begin{pmatrix} \mathbf{0} \\ \mathbf{0} \end{pmatrix}, \begin{pmatrix} \mathbf{\Sigma} & \mathbf{\Sigma} - \mathrm{diag}\{\mathbf{s}\} \\ \mathbf{\Sigma} - \mathrm{diag}\{\mathbf{s}\} & \mathbf{\Sigma} \end{pmatrix}\right). \tag{2}$$

Intuitively, a larger $\mathbf{s}$ implies that the constructed knockoff features are more different from the original features and thus can increase the power of the method.

With the constructed knockoff features $\tilde{\mathbf{x}}$, we quantify important features via the knockoff by resorting to the knockoff statistics $W_j = g_j(Z_j, \tilde{Z}_j)$ for $1 \leq j \leq p$, where $Z_j$ and $\tilde{Z}_j$ represent feature importance measures for the $j$th feature $X_j$ and its knockoff counterpart $\tilde{X}_j$, respectively, and $g_j(\cdot, \cdot)$ is an antisymmetric function satisfying $g_j(Z_j, \tilde{Z}_j) = -g_j(\tilde{Z}_j, Z_j)$. Note that the feature importance measures as well as the knockoff statistics depend on the specific algorithm used to fit the model. For example, in linear regression models one can choose $Z_j$ and $\tilde{Z}_j$ as the Lasso regression coefficients of $X_j$ and $\tilde{X}_j$, respectively, and a valid knockoff statistic could be $W_j = |Z_j| - |\tilde{Z}_j|$. In principle, the knockoff statistics $W_j$ should satisfy a coin-flip property such that swapping an arbitrary pair of $X_j$ and its knockoff counterpart $\widetilde{X}_j$ only changes the sign of $W_j$ but keeps the signs of other $W_k$ ($k \neq j$) unchanged [10]. A desirable property for knockoff statistics $W_j$'s is that important features are expected to have large positive values, whereas unimportant ones should have small magnitudes symmetric around 0.

Given the knockoff statistics as feature importance measures, we sort $|W_j|$'s in decreasing order and select features whose $W_j$'s exceed some threshold $T$. In particular, two choices of threshold are suggested [2, 10]

$$T = \min\left\{t \in \mathcal{W}, \frac{|\{j : W_j \leq -t\}|}{|\{j : W_j \geq t\}|} \leq q\right\}, \quad T_+ = \min\left\{t \in \mathcal{W}, \frac{1 + |\{j : W_j \leq -t\}|}{1 \vee |\{j : W_j \geq t\}|} \leq q\right\}, \tag{3}$$

where $\mathcal{W} = \{|W_j| : 1 \leq j \leq p\} \setminus \{0\}$ is the set of unique nonzero values attained by $|W_j|$'s and $q \in (0, 1)$ is the desired FDR level specified by the user. [10] proved that under mild conditions on $W_j$'s, the knockoff filter with $T$ controls a slightly modified version of FDR and the knockoff$_+$ filter with $T_+$ controls the exact FDR.

When the joint distribution of $\mathbf{x}$ is unknown, to construct knockoff features one needs to estimate this distribution from data. For the case of Gaussian design, the *approximate knockoff features* can be constructed by replacing $\mathbf{\Sigma}^{-1}$ in Equation 1 with the estimated precision matrix $\widehat{\mathbf{\Omega}}$. Following [18], we exploited the ISEE [21] for scalable precision matrix estimation when implementing DeepPINK.

## 3 Knockoffs inference for deep neural networks

In this paper, we integrate the idea of knockoff filters with DNNs to achieve feature selection with controlled FDR. We restrict ourselves to a Gaussian design so that the knockoff features can be constructed easily by following Equation 1. In practice, [10] shows how to generate knockoff variables from a *general* joint distribution of covariates.

We initially experimented with the multilayer preceptron by naively feeding the augmented feature vector $(\mathbf{x}^T, \tilde{\mathbf{x}}^T)^T$ directly into the networks. However, we discovered that although the FDR is controlled at the target level $q$, the power is extremely low and even 0 in many scenarios; see Table 1 and the discussions in a later section. To resolve the power issue, we propose a new flexible framework named DeepPINK for reproducible feature selection in DNNs, as illustrated in Figure 1.

The main idea of DeepPINK is to feed the network through a plugin pairwise-coupling layer containing $p$ filters, $F_1, \cdots, F_p$, where the $j$th filter connects feature $X_j$ and its knockoff counterpart $\tilde{X}_j$. Initialized equally, the corresponding filter weights $Z_j$ and $\tilde{Z}_j$ compete against each other during training. Thus intuitively, $Z_j$ being much larger than $\tilde{Z}_j$ in magnitude provides some evidence that the $j$th feature is important, whereas similar values of $Z_j$ and $\tilde{Z}_j$ indicate that the $j$th feature is not important.

In addition to the competition of each feature against its own knockoff counterpart, features compete against each other. To encourage competitions, we use a linear activation function in the pairwise-coupling layer.

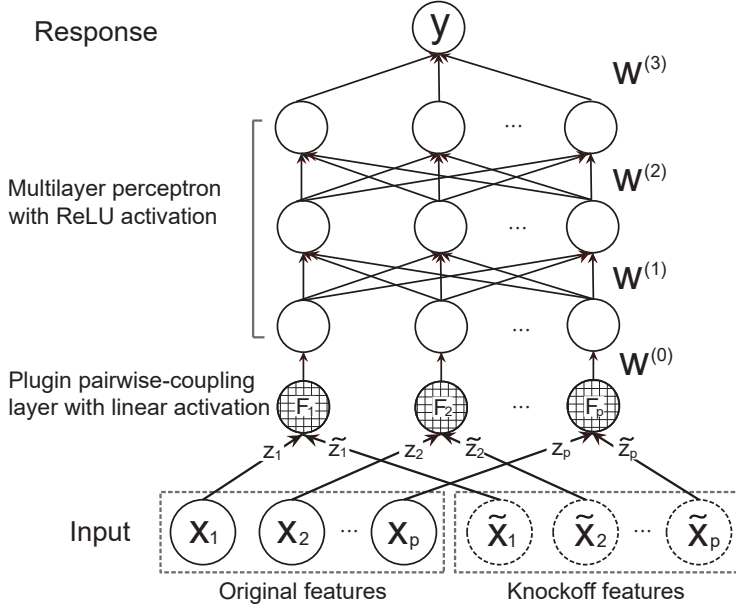

Response

Multilayer perceptron
with ReLU activation

Plugin pairwise-coupling
layer with linear activation

Input

Original features    Knockoff features

Figure 1: A graphical illustration of DeepPINK. DeepPINK is built upon an MLP with a plugin pairwise-coupling layer containing $p$ filters, one per input feature, where each filter connects the original feature and its knockoff counterpart. The filter weights $Z_j$ and $\tilde{Z}_j$ for $j$th feature and its knockoff counterpart are initialized equally for fair competition. The outputs of the filters are fed into a fully connected MLP with 2 hidden layers, each containing $p$ neurons. Both ReLU activation and $L_1$-regularization are used in the MLP.

The outputs of the filters are then fed into a fully connected multilayer perceptron (MLP) to learn a mapping to the response $Y$. The MLP network has multiple alternating linear transformation and nonlinear activation layers. Each layer learns a mapping from its input to a hidden space, and the last layer learns a mapping directly from the hidden space to the response $Y$. In this work, we use an MLP with 2 hidden layers, each containing $p$ neurons, as illustrated in Figure 1. We use $L_1$-regularization in the MLP with regularization parameter set to $O(\sqrt{\frac{2\log p}{n}})$. We use Adam [24] to train the deep learning model with respect to the mean squared error loss, using an initial learning rate of $0.001$ and batch size $10$.

Let $W^{(0)} \in \mathbb{R}^{p \times 1}$ be the weight vector connecting the filters to the MLP. And let $W^{(1)} \in \mathbb{R}^{p \times p}$, $W^{(2)} \in \mathbb{R}^{p \times p}$, and $W^{(3)} \in \mathbb{R}^{p \times 1}$ be the weight matrices connecting the input vector to the hidden layer, hidden layer to hidden layer, and hidden layer to the output, respectively.

The importance measures $Z_j$ and $\tilde{Z}_j$ are determined by two factors: (1) the relative importance between $X_j$ and its knockoff counterpart $\tilde{X}_j$, encoded by filter weights $\mathbf{z} = (z_1, \cdots, z_p)^T$ and $\tilde{\mathbf{z}} = (\tilde{z}_1, \cdots, \tilde{z}_p)^T$, respectively; and (2) the relative importance of the $j$th feature among all $p$ features, encoded by the weight matrices, $\mathbf{w} = W^{(0)} \odot (W^{(1)}W^{(2)}W^{(3)})$, where $\odot$ denotes the entry-wise matrix multiplication. Therefore, we define $Z_j$ and $\tilde{Z}_j$ as

$$Z_j = z_j \times \mathbf{w}_j \quad \text{and} \quad \tilde{Z}_j = \tilde{z}_j \times \mathbf{w}_j. \tag{4}$$

With the above introduced importance measures $Z_j$ and $\tilde{Z}_j$, the knockoff statistic can be defined as $W_j = Z_j^2 - \tilde{Z}_j^2$, and the filtering step can be applied to the $W_j$'s to select features. Our definition of feature importance measures naturally applies to deep neural nets with more hidden layers. The choice of 2 hidden layers for the MLP is only for illustration purposes.

## 4 Simulation studies

### 4.1 Model setups and simulation settings

We use synthetic data to compare the performance of DeepPINK to existing methods in the literature. Since the original knockoff filter [2] and the high-dimensional knockoff filter [3] were only designed for Gaussian linear regression models, we first simulate data from

$$\mathbf{y} = \mathbf{X}\boldsymbol{\beta} + \boldsymbol{\varepsilon}, \tag{5}$$

where $\mathbf{y} = (Y_1, \cdots, Y_n)^T \in \mathbb{R}^n$ is the response vector, $\mathbf{X} \in \mathbb{R}^{n \times p}$ is a random design matrix, $\boldsymbol{\beta} = (\beta_1, \beta_2, \cdots, \beta_p)^T \in \mathbb{R}^p$ is the coefficient vector, and $\boldsymbol{\varepsilon} = (\varepsilon_1, \cdots, \varepsilon_n)^T \in \mathbb{R}^n$ is a vector of noise.

Since nonlinear models are more pervasive than linear models in real applications, we also consider the following Single-Index model

$$Y_i = g(\mathbf{x}_i^T \boldsymbol{\beta}) + \varepsilon_i, \quad i = 1, \cdots, n, \tag{6}$$

where $g$ is some unknown link function and $\mathbf{x}_i$ is the feature vector corresponding to the $i$th observation.

We simulate the rows of $\mathbf{X}$ independently from $\mathcal{N}(\mathbf{0}, \boldsymbol{\Sigma})$ with precision matrix $\boldsymbol{\Sigma}^{-1} = \left(\rho^{|j-k|}\right)_{1 \leq j, k \leq p}$ with $\rho = 0.5$. The noise distribution is chosen as $\boldsymbol{\varepsilon} \sim \mathcal{N}(\mathbf{0}, \sigma^2 \mathbf{I}_n)$ with $\sigma = 1$. For all simulation examples, we set the target FDR level to $q = 0.2$.

For the linear model, we experiment with sample size $n = 1000$, and consider the high-dimensional scenario with number of features $p = 50, 100, 200, 400, 600, 800, 1000, 1500, 2000, 2500$, and $3000$. The true regression coefficient vector $\boldsymbol{\beta}_0 \in \mathbb{R}^p$ is sparse with $s = 30$ nonzero signals randomly located, and the nonzero coefficients are randomly chosen from $\{\pm 1.5\}$.

For the Single-Index model, we fix the sample size $n = 1000$ and vary feature dimensionality as $p = 50, 100, 200, 400, 600, 800, 1000, 1500, 2000, 2500$, and $3000$. We set the true link function $g(x) = x^3/2$. The true regression coefficient vector is generated similarly with $s = 10$.

## 4.2 Simulation results

We compare the performance of DeepPINK with three popularly used methods, MLP, DeepLIFT [34], random forest (RF) [9], and support vector regression (SVR) with linear kernel. For RF, the feature importance is measured by Gini importance [8]. For MLP, the feature importance is measured similarly to DeepPINK without the pairwise-coupling layer. For DeepLIFT, the feature importance is measured by the multiplier score. For SVR with linear kernel, the feature importance is measured by the coefficients in the primal problem [11]. We did not compare with SVR with nonlinear kernel because we could not find any feature importance measure with nonlinear kernel in existing software packages. We would like to point out that with the linear kernel, the model is misspecified when observations are generated from the Single-Index model. Due to the very high computational cost for various methods in high dimensions, for each simulation example, we set the number of repetitions to 20. For DeepPINK and MLP, the neural networks are trained 5 times with different random seeds within each repetition. When implementing the knockoff filter, the threshold $T_+$ was used, and the approximate knockoff features were constructed using the estimated precision matrix $\widehat{\boldsymbol{\Omega}}$ (see [18] for more details).

The empirical FDR and power are reported in Table 1. We see that DeepPINK consistently control FDR much below the target level even though we choose a pretty loose threshold and has the highest power among the competing methods in almost all settings. MLP, DeepLIFT, and RF control FDR at the target level, but in each case the power is very sensitive to dimensionality and model nonlinearity. For SVR with the linear kernel, occasionally the FDR is above the target level $q = 0.2$, which could be caused by the very small number of repetitions. It is worth mentioning that the superiority of DeepPINK over MLP and DeepLIFT lies in the pairwise-coupling layer, which encourages the importance competition between original and knockoff features directly. The results on FDR control are consistent with the general theory of knockoffs inference in [10] and [18]. It is worth mentioning that DeepPINK uses the same network architecture across different model settings.

## 5 Real data analysis

In addition to the simulation examples presented in Section 4, we also demonstrate the practical utility of DeepPINK on two real applications. For all studies the target FDR level is set to $q = 0.2$.

### 5.1 Application to HIV-1 data

We first apply DeepPINK to the task of identifying mutations associated with drug resistance in HIV-1 [32]. Separate data sets are available for resistance against different drugs from three classes: 7

| | DeepPINK | | | | MLP | | | | DeepLIFT | | | | RF | | | | SVR | | | |
|---|---|---|---|---|---|---|---|---|---|---|---|---|---|---|---|---|---|---|---|---|
| | Linear | | Single-Index | | Linear | | Single-Index | | Linear | | Single-Index | | Linear | | Single-Index | | Linear | | Single-Index | |
| $p$ | FDR | Power | FDR | Power | FDR | Power | FDR | Power | FDR | Power | FDR | Power | FDR | Power | FDR | Power | FDR | Power | FDR | Power |
| 50 | 0.046 | 1 | 0.13 | 0.98 | 0.15 | 1 | 0.17 | 0.89 | 0.16 | 1 | 0.24 | 0.9 | 0.005 | 0.45 | 0 | 0 | 0.18 | 1 | 0.180 | 0.81 |
| 100 | 0.047 | 1 | 0.08 | 1 | 0.048 | 1 | 0.056 | 0.26 | 0.16 | 1 | 0.13 | 0.47 | 0.016 | 0.61 | 0.025 | 0.045 | 0.22 | 1 | 0.094 | 0.26 |
| 200 | 0.042 | 0.99 | 0.042 | 1 | 0.11 | 1 | 0 | 0 | 0.24 | 0.96 | 0.034 | 0.067 | 0.013 | 0.54 | 0.020 | 0.045 | 0.21 | 1 | 0.061 | 0.05 |
| 400 | 0.022 | 0.97 | 0.022 | 1 | 0.29 | 0.95 | 0 | 0 | 0.034 | 0.5 | 0.039 | 0.069 | 0.017 | 0.53 | 0.033 | 0.050 | 0.22 | 1 | 0.083 | 0.01 |
| 600 | 0.031 | 0.95 | 0.046 | 1 | 0.17 | 0.8 | 0.014 | 0.013 | 0.003 | 0.26 | 0.068 | 0.16 | 0.023 | 0.56 | 0.11 | 0.095 | 0.19 | 1 | 0 | 0 |
| 800 | 0.048 | 0.95 | 0.082 | 1 | 0.037 | 0.62 | 0.016 | 0.068 | 0 | 0.17 | 0.16 | 0.24 | 0.022 | 0.61 | 0.061 | 0.12 | 0.22 | 0.98 | 0 | 0 |
| 1000 | 0.023 | 0.97 | 0.065 | 1 | 0.007 | 0.4 | 0.037 | 0.16 | 0 | 0.12 | 0.013 | 0.33 | 0.029 | 0.59 | 0.081 | 0.17 | 0.15 | 0.67 | 0 | 0 |
| 1500 | 0.007 | 1 | 0.065 | 1 | 0.002 | 0.41 | 0.068 | 0.25 | 0.001 | 0.32 | 0.13 | 0.44 | 0.045 | 0.58 | 0.098 | 0.17 | 0.064 | 0.043 | 0 | 0 |
| 2000 | 0.026 | 0.99 | 0.098 | 1 | 0.023 | 0.4 | 0.063 | 0.35 | 0.015 | 0.37 | 0.1 | 0.56 | 0.033 | 0.65 | 0.046 | 0.14 | 0.04 | 0.002 | 0 | 0 |
| 2500 | 0.029 | 0.97 | 0.067 | 1 | 0.21 | 0.5 | 0.042 | 0.35 | 0.088 | 0.58 | 0.32 | 0.47 | 0.034 | 0.62 | 0.11 | 0.18 | 0.02 | 0.005 | 0 | 0 |
| 3000 | 0.046 | 0.97 | 0.051 | 1 | 0.11 | 0.43 | 0.046 | 0.31 | 0.069 | 0.46 | 0.14 | 0.44 | 0.05 | 0.65 | 0.087 | 0.17 | 0.05 | 0 | 0 | 0 |

Table 1: Simulation results for linear model and the Single-Index model.

protease inhibitors (PIs), 6 nucleoside reverse-transcriptase inhibitors (NRTIs), and 3 nonnucleoside reverse transcriptase inhibitors (NNRTIs). The response $Y$ is the log-transformed drug resistance level to the drug, and the $j$th column of the design matrix $\mathbf{X}$ indicates the presence or absence of the $j$th mutation.

We compare the identified mutations, within each drug class, against the treatment-selected mutations (TSM), which contains mutations associated with treatment by the drug from that class. Following [2], for each drug class, we consider the $j$th mutation as a discovery for that drug class as long as it is selected for any of the drugs in that class. Before running the selection procedure, we remove patients with missing drug resistance information and only keep those mutations which appear at least three times in the sample. We then apply three different methods to the data set: DeepPINK with model-X knockoffs discussed earlier in this paper, the original fixed-X knockoff filter based on a Gaussian linear model (Knockoff) proposed in [2], and the Benjamini–Hochberg (BHq) procedure [4]. Following [2], z-scores rather than p-values are used in BHq to facilitate comparison with other methods.

Figure 2 summarizes the discovered mutations of all methods within each drug class for PI and NRTI. In this experiment, we see several differences among the methods. Compared to Knockoff, DeepPINK obtains equivalent or better power in 9 out of 13 cases with comparable false discovery proportion. In particular, DeepPINK and BHq are the only methods that can identify mutations in DDI, TDF, and X3TC for NRTI. Compared to BHq, DeepPINK obtains equivalent or better power in 10 out of 13 cases with with much better controlled false discovery proportion. In particular, DeepPINK shows remarkable performance for APV where it recovers 18 mutations without any mutation falling outside the TSM list. Finally, it is worth mentioning that DeepPINK does not make assumptions about the underlying models.

## 5.2 Application to gut microbiome data

We next apply DeepPINK to the task of identifying the important nutrient intake as well as bacteria genera in the gut that are associated with body-mass index (BMI). We use a cross-sectional study of $n = 98$ healthy volunteers to investigate the dietary effect on the human gut microbiome [12, 26, 45]. The nutrient intake consists of $p_1 = 214$ micronutrients whose values are first normalized using the residual method to adjust for caloric intake and then standardized [12]. Furthermore, the composition of $p_2 = 87$ genera are extracted using 16S rRNA sequencing from stool samples. the compositional data is first log-ratio transformed to get rid of the sum-to-one constraint and then centralized. Following [26], 0s are replaced with 0.5 before converting the data to compositional form. Therefore, we treat BMI as the response and the nutrient intake together with genera composition as predictors.

Table 2 shows the eight identified nutrient intake and bacteria genera using DeepPINK with the target FDR level $q = 0.2$. Among them, three overlap with the four genera (*Acidaminococcus*, *Alistipes*, *Allisonella*, *Clostridium*) identified in [26] using Lasso. At the phylum level, it is known that firmicutes affect human obesity [26], which is consistent with identified firmicutes-dominated genera. In view of the associations with BMI, all identified genera and nutrient intake by DeepPINK are supported by literature evidence shown in Table 2.

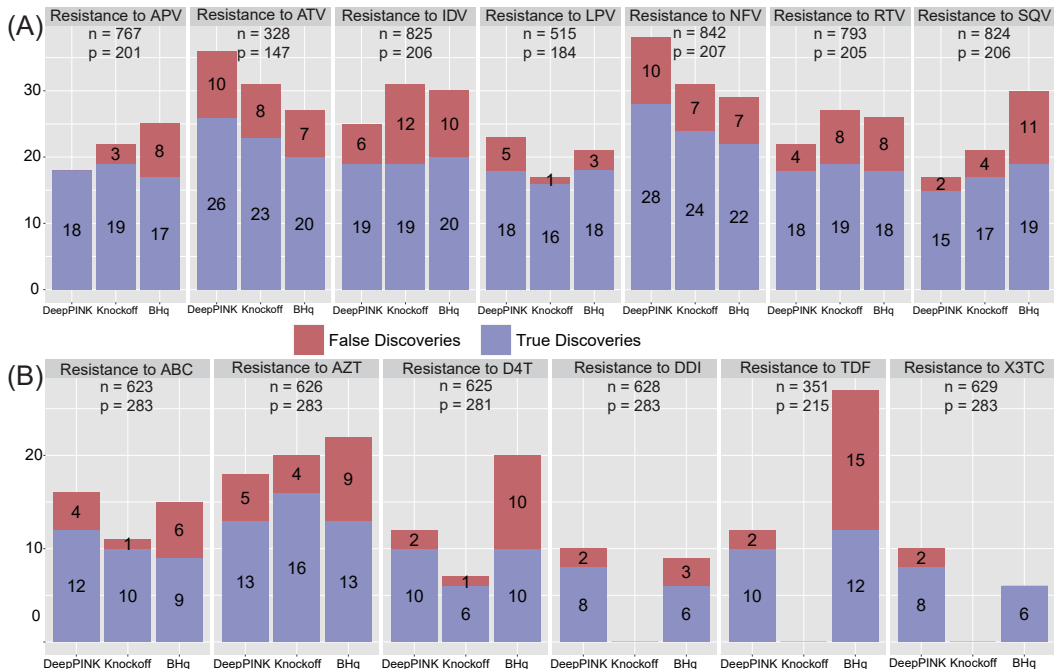

Figure 2: Performance on the HIV-1 drug resistance data. For each drug class, we show the number of mutation positions for (A)PI and (B)NRTI identified by DeepPINK, Knockoff, and BHq at target FDR level $q = 0.2$.

| | Nutrient intake | | | Bacteria genera | | |
|---|---|---|---|---|---|---|
| | Micronutrient | Reference | | Phylum | Genus | Reference |
| 1 | Linoleic | [7] | | Firmicutes | Clostridium | [26] |
| 2 | Dairy Protein | [29] | | Firmicutes | Acidaminococcus | [26] |
| 3 | Choline, Phosphatidylcholine | [31] | | Firmicutes | Allisonella | [26] |
| 4 | Choline, Phosphatidylcholine w/o suppl. | [31] | | Firmicutes | Megamonas | [25] |
| 5 | Omega 6 | [39] | | Firmicutes | Megasphaera | [43] |
| 6 | Phenylalanine, Aspartame | [41] | | Firmicutes | Mitsuokella | [43] |
| 7 | Aspartic Acid, Aspartame | [41] | | Firmicutes | Holdemania | [30] |
| 8 | Theaflavin 3-gallate, flavan-3-ol(2) | [42] | | Proteobacteria | Sutterella | [13] |

Table 2: Major nutrient intake and gut microbiome genera identified by DeepPINK with supported literature evidence.

# 6  Conclusion

We have introduced a new method, DeepPINK, that can help to interpret a deep neural network model by identifying a subset of relevant input features, subject to the FDR control. DeepPINK employs a particular DNN architecture, namely, a plugin pairwise-coupling layer, to encourage competitions between each original feature and its knockoff counterpart. DeepPINK achieves FDR control with much higher power than the naive combination of the knockoffs idea with a vanilla MLP. Extending DeepPINK to other neural networks such as CNN and RNN would be interesting directions for future.

Reciprocally, DeepPINK in turn improves on the original knockoff inference framework. It is worth mentioning that feature selection always relies on some importance measures, explicitly or implicitly (e.g., Gini importance [8] in random forest). One drawback is that the feature importance measures heavily depend on the specific algorithm used to fit the model. Given the universal approximation property of DNNs, practitioners can now avoid having to handcraft feature importance measures with the help of DeepPINK.

**Acknowledgments**

This work was supported by NIH awards 1R01GM131407-01 and R01GM121818, a grant from the Simons Foundation, and an Adobe Data Science Research Award.

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
