[Reviews · NeurIPS 2018]

Reviewer 1



Summary This paper proposes a new method called, DeepPINK, for hypothesis testing on deep learning (DL) models. Specifically, knockoff filters are seamlessly incorporated into the DL framework. Superior power and FDR control are shown over existing methods for both synthetic and real data. Quality The paper is technically sound with both theoretical and experimental support. It would be nice to see comparisons against some existing methods, such as DeepLIFT, without applying knockoff filtering, even if those methods provide no statistical guarantees. Clarity The paper is clearly written and well organized. Great motivation is provided in the Introduction. The only part that is unclear from the text is whether building knockoffs requires seeing Y. Originality The idea of knockoff filters is not new, but the DL architecture required to exploit this idea, as proposed in this paper, is new and shown to be critical. Significance The problem of interpreting DL model is extremely difficult. DeepPINK is by far the best solution I have seen with statistical guarantees on the selected features. I expect this paper to inspire many other researchers. After rebuttal The authors have addressed my remaining concerns, but I can't increase the score any higher since I already gave it a 10.

Reviewer 2



The paper proposes a method for feature selection in neural networks using a method for a controlled error rate, quantified through the False Discovery Rate. To control FDR the paper is using the model-X knockoffs framework [2, 3, 10]: construct random features that obey distributional properties with respect to the true features, and extract statistics (filters) of pairwise importance measures between true-knockoff dimensions. The choice of the importance function and the knockoff filters is flexible. The novelty of this paper lies in using a neural network (MLP) to get the importance measure through a linear layer that couples the true and knockoff features pairwise. The final statistic depends both on the trainable linear layer weights and the rest of the network weights. Results are provided on real and synthetic data: for the latter (noisy linear and nonlinear processes with increasing number of dimensions/features), the proposed method performs better (in terms of FDR control and power or feature relevance certainty) compared to MLP, RF and SVMs. Notably, compared to MLP the difference comes from the coupling initial, linear layer. The paper also compares feature selection to other knockoffs-related methods ([2], [4], [10]), on a real dataset for mining interpretable dimensions features (HIV-1) and provide qualitative analysis on the discovered features from gut microbiome data. General comments: Feature selection is an important topic related to interpretability, transparency and robustness in neural network models (and machine learning more general) and the paper is an effort in that direction. The framework is an extension of the model-X knockoff framework [10] and the novelty lies in adding a hard-wired, linear, pairwise-connected layer on the knockoff features before an MLP. The benefit is higher statistical power for feature relevance. I like the idea of coupling feature weights and algorithm weights in a single importance measure, which ends up ranking a feature j high if both the filter coefficients difference and the NN output dimension j is high. The paper has a clear exposure of ideas and background (e.g. on the knockoff statistics models provided) and the paper is easy to follow from motivation, to background to empirical evaluation. My concerns with the paper are mainly around: a) the amount of novelty (and generality) on the proposed coupling/pairwise layer and b) the broader applicability of the proposed method. Strengths: - The framework is an application/extension of an existing statistical framework for error control with theoretical guarantees [10]. - The coupling pairs + parametrization idea is generic and could be applied to different types of algorithms (as a learned representation layer). - The paper is well written, reads smoothly and the ideas are well exposed (supported, explained or cited). Weaknesses: - It would help to have some element of theoretical analysis on the chosen definition of the importance measures Z (Sec. 3). For example, deriving this as an explicit regularizer on the layer weights. - The framework (proposed as a general feature selection method?), is restricted to Gaussian design and the use of an estimated precision matrix, for which there might not be any robustness guarantees. Some discussion (or paths for alternative designs?) might be helpful. - There is some disconnect, in my opinion, between motivating this work as a feature selection method for NN and using a NN as a way to measure importance (given a prediction task). - The empirical validation, even though convincing in some respect: the benefit of the novelty here (i.e. adding a pairwise connected layer vs. having a vanilla MLP) is only explored on the synthetic data. Specific comments: - The method is motivated by the need for interoperability in NN and error control in NN, though only the latter is explored (in a synthetic scenario). - Similarly, the title states “reproducible” feature selection and this might something that needs further explaining. In what sense reproducible? - Did the authors explore/consider different definitions of the importance measures on the knockoff statistics? - What is the dependency of some aspects of the performance on the knock-off feature generation (e.g., s for the Gaussian case)? Is there a way to make vanilla MLP + knockoff features better by changing (or tuning) s? - Is there a specific rationale for fixing the number of neurons in the hidden layer to be p (equal to the input dimension)? As long as the output layer is p-dimensional the reasoning should still hold with a different ‘factorization’ of the input to output mapping that defines w. The same holds for using L1 regularization (and with a specific regularization parameter). - How dependent on the specific parametrization of the MLP (or NN more general) is the performance? - The FDR target error level is fixed/set to q=0.2 everywhere. Would it be insightful to vary this and re-interpret, e.g. the results of Sec. 5.1? - It would be helpful to see some prediction results with real data, based on the selected features. E.g. a two-pass process where the NN is re-trained on the selected, error-control features to predict the target function values or class. - How much False Discovery errors are relevant for NN performance in practical applications (and in high-dimensions)? An open question that might be worth discussing. - Did the authors try the knockoffs+vanilla MLP approach on the real data in 5.1 (or the qualitative analysis in 5.2)? The pairwise-connected layer is the novelty of this paper and this should be further highlighted. ========================= Comments after author feedback: Thank you for the replies to all reviews and most of the comments. I agree with the points of your clarifications and can see the value of the points made in the response. I also agree with Reviewer 1 that this can inspire future work by the authors and others. I am changing my score to a 6 to reflect these. At the same time my main concern on general applicability remains, and in some cases has been strengthened by some of the responses. - “in the sense that small changes in the data can yield dramatic changes in the selected features. “ If that is the case, perturbations in the data (and thus reproducibility of the framework) are not explored. What type of changes? - “We consider the primary virtue of the robustness to be improved reproducibility of the selected features;” As above. I don’t feel that this aspect of the authors’ work is highlighted or supported experimentally in the paper. - “Given the universal approximation property of DNNs, practitioners can now avoid having to handcraft feature importance measures with the help of DeepPINK.” This is a good point, thank you for making this connection explicit. Is the addition of DeepPINK layer however ‘altering’ the network definition, i.e. are you selecting features for the original network or the modified one? How dependent is this selection on initialization, parametrization, and as the authors state, changes in the input data (e.g. transformations, subsampling, noise etc). - “using interpretability alone as a measurement is far less reliable than using FDR and power with synthetic data as measurements.” This is also a good point, and provides statistical guarantees (as Reviewer 1 also points out). - “it is the fundamental architecture (instead of the tuning) of vanilla MLP that prevents the knockoff filter from performing well.” I am not sure what this claim implies. That MLP is not a good choice for being subjected to knockoff filters? This defies the purpose of the paper. Or that knockoffs can only add value coupled in the learning process (like for DeepPINK)? I wonder, and I made the same point in my initial review on which the authors replied, how much those findings are generally applicable to any ‘neural network’ parametrization or just MLP-type. How about basic linear or (shallow) non-linear regression with the PINK coupling and solved through SGD? - “Since there is no existing method available for robust feature selection in DNNs, any effective methods would be significant.” I agree. However the idea of input features and feature representation (hidden layers) is coupled in NNs. As the authors point out, interpretability and selection is conditioned on the prediction algorithm. So this is not ‘feature selection’, but input dimension selection. One could envision however a PINK layer attached on higher hidden layers for representation dimension selection, interpretability and transferability. - “The proposed network architecture may seem simple, but our journey to finding it was nontrivial.” So then does the paper involve a single, fixed architecture? Or is it generally applicable for any parametrization that one can add on top of the pairwise linear layer?

Reviewer 3



The idea of the paper is to use FDR and the knockoff filter for feature selection in neural networks. Of course both are already proposed by other work. I have the following concern of this paper. 1. First of all, I'm a bit skeptical about the originality of the paper. The paper is more like an application of existing work to neural networks. 2. The most serious point is I was unable to find why this approach is good for neural networks. My impression is the authors were using some relevant approach long time and so now just applying them to neural networks, finding one way of generating (or mimicking) the knockoff filter in neural networks. In other words I was unable to see any clear reason why the method is good for neural networks. I do not think this way is a proper motivation for developing a feature selection method for neural networks. 3. The method is proposed for feature selection. So the authors should compare their method with other feature selection methods more intensively. Honestly exploring this direction further, the comparison might be just between FDR (with knokoff filter) and other approaches. This is the reason I commented in #1.